# Impact of Mixing Shear on Polymer Binder Molecular Weight and Battery Electrode Reproducibility

**Samantha L. Morelly, Renee M. Saraka, Nicolas J. Alvarez *** and **Maureen Tang ***

Department of Chemical and Biological Engineering, Drexel University, Philadelphia, PA 19104, USA
* Correspondence: nja49@drexel.edu (N.J.A.); mhtang@drexel.edu (M.T.)

**Abstract:** The viscosity and microstructure of Li-ion battery slurries and the performance of the resulting electrodes have been shown to depend on the mixing protocol. This work applies rheology to understand the impact of shear during mixing and polymer molecular weight on slurry microstructure and electrode performance. Mixing protocols of different shear intensity are applied to slurries of $LiNi_{0.33}Mn_{0.33}Co_{0.33}O_2$ (NMC), carbon black (CB), and polyvinyldiene difluoride (PVDF) in N-methyl-2-pyrrolidinone (NMP), using both high-molecular-weight (HMW) and low-molecular-weight (LMW) PVDF. Slurries of both polymers are observed to form colloidal gels under high-shear mixing, even though unfavorable interactions between high molecular weight PVDF and CB should prevent this microstructure from forming. Theoretical analysis and experimental results show that increasing shear rate during the polymer and particle mixing steps causes polymer scission to decrease the polymer molecular weight and allow colloidal gelation. In general, electrodes made from high molecular weight PVDF generally show increased rate capability. However, high shear rates lead to increased cell variability, possibly due to the heterogeneities introduced by polymer scission.

**Keywords:** rheology; battery manufacturing; colloidal suspension

## 1. Introduction

Optimal processing of multi-component composite electrodes is essential for high-performance Li-ion and other advanced batteries. Applying tools from rheology, the study of fluid flow and deformation, can provide insight into how the electrode microstructure is influenced by the mixing, coating, and drying operation and how those changes in microstructure impact performance. Despite this research effort, much of the knowledge for obtaining a homogeneous electrode is highly empirical and system-specific. Developing principles that can be applied across systems for producing optimal electrodes requires better understanding of the phenomena in electrode processing.

Mixing a slurry of active material, conductive carbon additive, and polymer binder is the first step of battery manufacturing. Mixing of battery slurries can be generally divided into two processes: (1) dry-mixing and (2) wet-mixing. Dry-mixing is used to homogenize the active material and conductive additives, which are often of different sizes. At higher dry-mixing intensities, the conductive carbon adheres to the surface of the active material. The wet-mixing step is used to disperse the active material and conductive additive with the polymer binder and solvent. Despite several decades of research and development, there are very few general guidelines in literature of the "best" method, order and intensity at which both dry- and wet-mixing steps should be implemented. The reason behind this confusion is the complex relationship between conductive carbon, polymer binder, and the physical environments they experience during manufacturing. Entwistle et al. recently reviewed the complexity of this process [1].

A combination of observations from literature have demonstrated apparent history-dependent behavior during wet-mixing, dry-mixing, or a combination of the processes. In one study, Kim et al. showed the impact of four different mixing procedures with varying

complexity on slurry viscosity and electrode performance for $LiCoO_2$ cathodes. They found that the viscosity from each of these procedures was different and required different amounts of wet-mixing time to achieve a homogeneous slurry. Electrodes made from these different slurries showed varying amounts of capacity fade [2]. Lee et al. studied cathode slurries made with $LiCoO_2$, carbon black (CB), and polyvinyldiene difluoride (PVDF) and showed how changing the number of wet-mixing steps impacted the battery slurry microstructure and electrode performance. They found that increasing the number of mixing steps changed the slurry microstructure from a gel to a viscoelastic fluid. Electrodes made from the viscoelastic fluid had higher capacities, lower resistance and less capacity fade over time [3]. Bockholt et al. studied the impact of dry-mixing on cathode performance and found different rate performances for different mixing apparatuses despite using the same overall procedure [4–6]. Changes in electrode performance with different dry-mixing equipment can be explained if some types of equipment break up conductive additive agglomerates more effectively than others for improved contact between conductive additive and active material [7]. However, measured electrode differences arising from changes in wet-mixing are not as easily explained.

Understanding the influence of battery slurry processing history on final electrode performance requires fundamental analysis from the field of colloidal science. Previous work by our group found that battery slurries of $LiNi_{0.33}Mn_{0.33}Co_{0.33}O_2$ (NMC), CB, and PVDF can be classified as colloidal suspensions driven by the depletion interactions between CB aggregates and PVDF. The colloidal suspensions can form a gel at a critical volume fraction of CB, $\varphi_{CB} \sim 0.02$ [8]. Colloidal suspension stability and the critical volume fraction can be predicted from easily measured characteristics such as particle/aggregate size, density, polymer molecular weight (Mw), and concentration [9–11]. When there are interacting components in a colloidal suspension, the shear and temperature history of the suspension becomes very important. Several of the parameters that impact the behavior of colloidal suspensions, most notably the dominant aggregate size and the polymer Mw, are affected by the type and intensity of shear applied during mixing and the drying rate (drying temperature) [12]. Serra et al. studied this phenomenon and showed that increasing the shear stress applied during mixing decreases the average aggregate size observed after mixing [13]. Griessl et al. also found that mixing shear broadened the particle size distribution [14]. Aggregate size plays an important role in the gelation phenomenon. For example, the intra-particle interactions are a strong function of the CB aggregate size and will directly impact the critical volume fraction for gelation [11,15]. Applied shear rate can also impact gel formation. We recently found that low shear rates during battery slurry coating densify the electrode microstructure, while high shear rates increase gel network strength. Stronger networks with better connectivity had beneficial impacts on electron transport and battery performance [12].

Additionally, polymers are known to undergo chain scission, or reduction in Mw, in fast flows [16,17]. The Mw of the average chain also influences the intra-particle interaction energy and thus the critical gelation criteria. Overall, shear-induced changes in CB aggregate size and/or PVDF Mw could explain apparent history-dependence during battery slurry mixing. The possibility of changing polymer Mw during mixing introduces an additional question—how important is the Mw of the polymer to electrode performance? Yao et al. found that higher Mw was beneficial to performance, but attributed effects primarily to mechanical flexibility around silicon particles with large volume changes [18]. A study by Li et al. investigated how the Mw and concentration of polyethyleneimine impacted the agglomeration of $LiFePO_4$ and CB. They found that agglomeration of $LiFePO_4$ and CB was prevented by different Mws and concentrations of polymer [19]. Lee et al. studied the impact of Mw of carboxymethyl cellulose and the degree of substitution on the performance of $Li_4Ti_5O_{12}$ anodes. Their results found an optimal Mw that resulted in higher electrode performance and more desirable mechanical properties [20]. Byun et al. found that increasing PVDF Mw increased adhesion of the electrode to the aluminum current collector and the adhesion between particles in the electrode microstructure [21]. Furthermore,

there was a clear correlation between binder Mw, adhesion, and improved cycle lifetime at elevated temperatures. Byun et al.'s results have important implications for the work of Apachitei et al., which showed that the a given battery formulation can be optimized using multi-variate analysis if the adhesion effect of the binder is known [22]. These studies point towards the importance of the polymer Mw for both anode and cathode performance.

This work investigates the impact of shear during mixing on battery slurry rheology, electrode performance, and experimental reproducibility. NMC and CB with two different polymer Mws serve as a baseline cathode chemistry. We systematically vary the shear applied during both polymer-mixing and particle-mixing via the choice of mixing equipment (Figure 1) and interpret the resulting slurry microstructures with respect to colloidal theory and polymer physics. Finally, producing electrodes from the slurries and characterizing their electrochemical rate capability determines the role of shear during mixing and polymer Mw on battery performance.

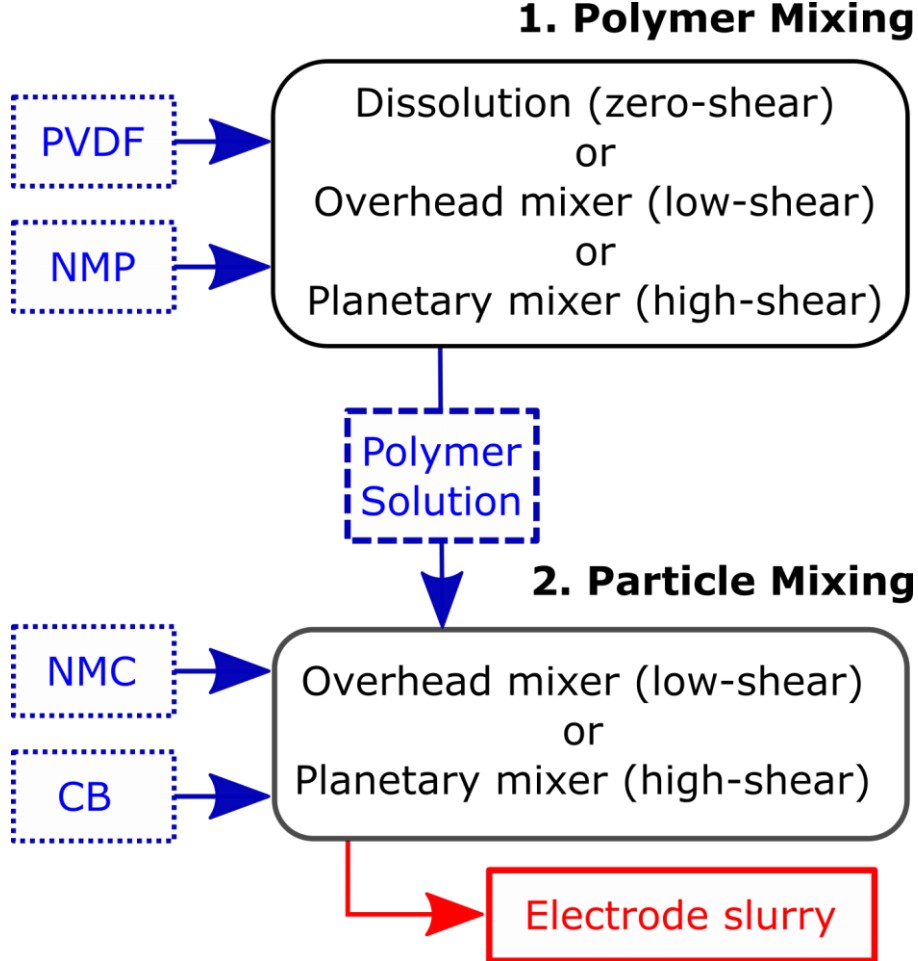

**Figure 1.** Mixing protocol used to make 12 battery slurries.

## 2. Materials and Methods

Materials and sample preparation: $LiNi_{0.33}Mn_{0.33}Co_{0.33}O_2$ (NMC, NM-3100, Toda America, Battle Creek, MI, USA) and carbon black (CB, Super C65, Timcal, Bodio, Switzerland) were used as received. Toda reports NMC's particle size to be 10 microns and CB has a reported particle size of 100 nm [23]. Low and high Mw polyvinylidene difluorides (LMW PVDF and HMW PVDF), were used as received from Arkema (King of Prussia, PA, USA). The LMW PVDF has a Mw = 380,000 g/mol (Kynar 301F) as reported by the manufacturer, while HMW PVDF, (Kynar HSV 900) was reported to be a blend of two Mws, $Mw_1$ = 92,840 kg/mol and $Mw_2$ = 1367 kg/mol as determined by size exclusion chromatog-

raphy [24]. 1-Methyl-2-pyrrolidinone, NMP, was used as the solvent (Sigma Aldrich, purity $\geq$ 99.0%, St. Louis, MO, USA). The electrode composition tested was 95wt% NMC, 2.5wt% CB, 2.5wt% PVDF, and the polymer concentration in solution was 48 mg/mL.

Slurries were mixed in accordance to the procedure shown in Figure 1. Dissolution was assisted by a Benchtop Roller (Wheaton, Millville, NJ, USA) and took 2–4 days. The overhead mixer (One Cell Systems, Inc., Cambridge, MA, USA) was used with a 1″ three-blade paddle. The planetary mixer was the ARE- 310 non-vacuum type planetary centrifugal mixer from Thinky USA (Laguna, CA, USA). For the polymer mixing step, the PVDF was mixed into NMP for 30 min or 10 min for the overhead and planetary mixers, respectively. For the particle mixing step, CB and NMC were added to the PVDF solution for 1.5 h or 10 min for the overhead and planetary mixers, respectively.

After mixing, slurries were coated onto an aluminum foil current collector (20 μm) with an automatic coater (TOB Energy) and a doctor blade set to 100 μm. Video recording calculated an applied coating shear rate of 200–300 1/s. The film was dried at room temperature overnight and under vacuum at 120 °C for 12 h. After drying, 0.375″ diameter electrodes were punched from the film. Individual electrodes were calendered at 20 MPa of pressure with a Carver melt press. Electrodes were heated for 1 h at 100 °C to remove water, then cycled into an Ar atmosphere glovebox (LP Technology Solutions). Electrodes were assembled into 2032 sized coin cells with LP30 electrolyte (Gotion), two Celgard separators, and 0.5″ diameter lithium counter electrodes.

Rheological Characterization: Oscillatory rheometry was performed on a DHR-3 rheometer (TA Instruments, Newcastle, DE, USA) using a 40 mm parallel plate geometry at $T$ = 25 °C with a Peltier plate setup. Pouring or gentle spatula movements were used to load samples onto the parallel plate. The geometry was lowered slowly to ensure that no air bubbles were entrained, and the parallel plate geometry minimized confinement effects. To determine gap effects, the linear viscoelastic measurements were measured at gap heights between h = 300 μm to 1 mm. The results found no gap effects at h $\geq$ 500 μm. Oscillatory strain sweeps at fixed angular frequency, $\omega$ = 1 rad/s and frequency sweeps at fixed strain amplitude, $\gamma$ = 0.003, were performed after waiting for 510 min or longer to ensure sample equilibration and that the normal force had returned to zero. Steady shear viscosity measurements were performed on an AR-2000 rheometer (TA Instruments, Newcastle, DE, USA) using a 40 mm cone and plate geometry at $T$ = 25 °C and a Peltier plate setup.

Battery Testing: Rate capability tests were performed using an Arbin battery cycler (Arbin Instruments, College Station, TX, USA). Batteries were first charged and discharged for four cycles from 4.3 to 3 V at C/10. Following cycles were charged at C/10 and discharged at C/2, 1C, 2C, 5C, 10C, and then another C/2 for four cycles at each rate.

## 3. Results and Discussion

### 3.1. Polymer Scission during Slurry Mixing

Figure 2 shows small angle oscillatory shear results for the twelve battery slurries studied. The left-hand column (a, c, e) shows the complex shear moduli for six mixing protocols with LMW PVDF, and the right-hand column (b, d, f) shows the moduli obtained with six mixing protocols with HMW PVDF. The colors represent different mixing procedures, as denoted by the bottom right legend. Filled symbols represent G′, the storage modulus, and hollow symbols represent G″, the loss modulus. For the discussion here, we classify slurries as strong gels, weak gels, or viscoelastic fluids. We define strong gels by storage and loss moduli that depend weakly on frequency and by passing an inversion test. Weak gels have a storage modulus that is independent of frequency, but a loss modulus that is frequency-dependent. Weak gels will also flow due to gravity and fail the inversion test. Viscoelastic fluids have moduli that depend strongly on the angular frequency, and a loss modulus that is greater than the storage modulus for the majority of the angular frequencies. The microstructures obtained are summarized in Table 1.

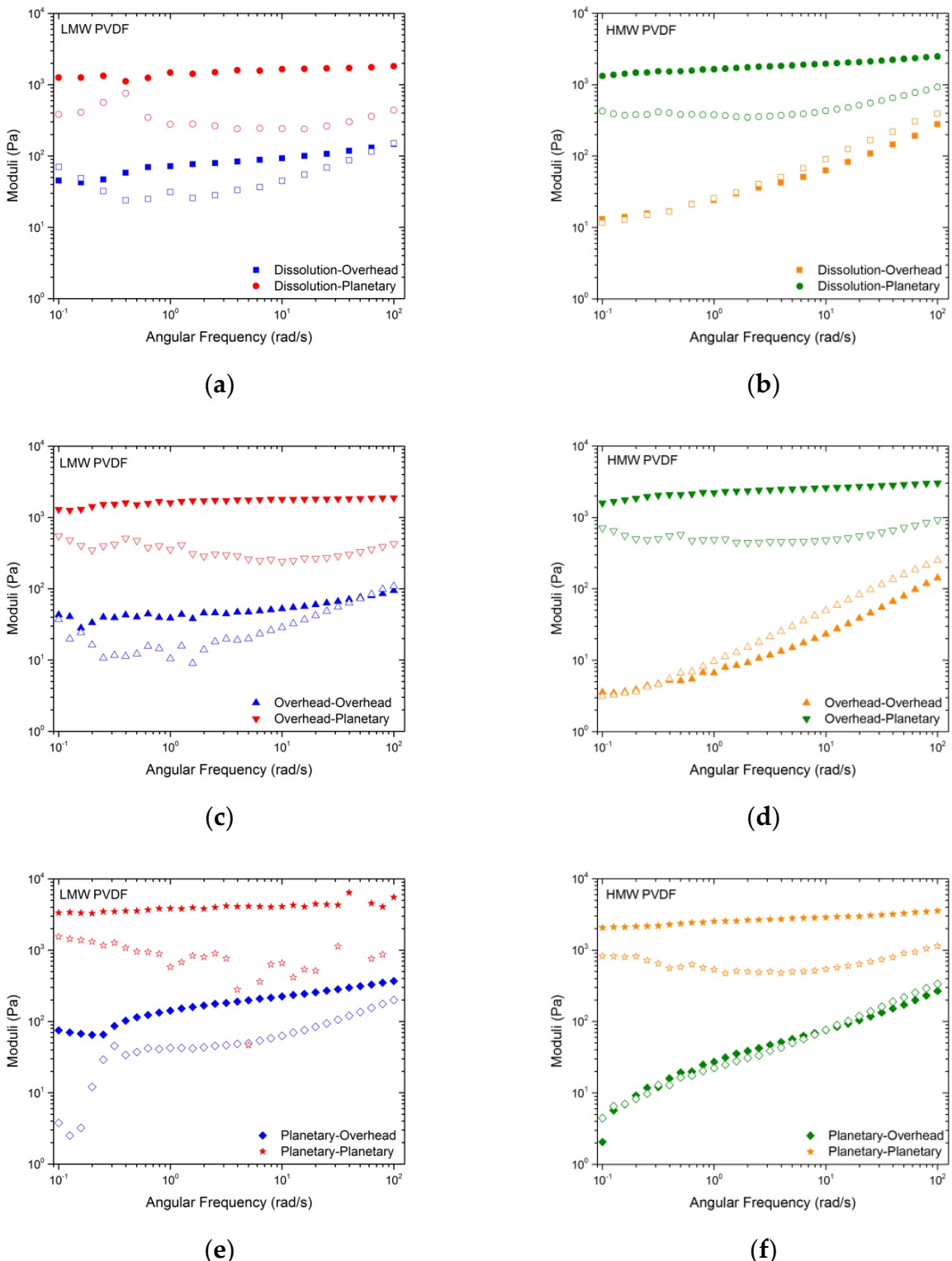

**Figure 2.** Small angle oscillatory shear for the two slurry formulations and six mixing protocols. Each column corresponds to a different polymer MW. (**a**,**c**,**e**) are LMW PVDF slurries while (**b**,**d**,**f**) are HMW PVDF slurries. Rows correspond to slurries made with the same polymer mixing step. (**a**,**b**) correspond to dissolution (zero−shear) polymer mixing, (**c**,**d**) to the overhead mixer (low−shear) and (**e**,**f**) to the planetary mixer (high−shear). The secondary mixing step is used to mix the particles into solution and is either via overhead or planetary mixer. Filled symbols represent G′, the storage modulus, and hollow symbols represent G″, the loss modulus.

**Table 1.** Summary of microstructures obtained from slurries mixed with either high or low Mw PVDF and various shear for polymer and particle mixing. The shear applied increases from dissolution (zero-shear), overhead mixer (low-shear), or the planetary mixer (high-shear).

| Figure | Polymer MW | Polymer Mixing | Particle Mixing | Microstructure |
|---|---|---|---|---|
| Figure 2a | LMW | Dissolution | Overhead | Weak gel |
| | LMW | Dissolution | Planetary | Strong gel |
| Figure 2b | HMW | Dissolution | Overhead | Fluid |
| | HMW | Dissolution | Planetary | Strong gel |
| Figure 2c | LMW | Overhead | Overhead | Weak gel |
| | LMW | Overhead | Planetary | Strong gel |
| Figure 2d | HMW | Overhead | Overhead | Fluid |
| | HMW | Overhead | Planetary | Strong gel |
| Figure 2e | LMW | Planetary | Overhead | Weak gel |
| | LMW | Planetary | Planetary | Strong gel |
| Figure 2f | HMW | Planetary | Overhead | Fluid |
| | HMW | Planetary | Planetary | Strong gel |

Depending on the mixing protocol, all three types of microstructures are identified. Specifically, LMW and HMW PVDF-based slurries typically result in a strong gel when the particle are mixed with the high-shear planetary mixer. LMW PVDF based samples utilizing an overhead mixer typically result in a weak gel. In contrast, HMW PVDF samples mixed using an overhead mixer yield viscoelastic fluids.

Our previous work has demonstrated effectively that Poon's theory for predicting the gel point of a colloidal suspension is valid for multi-component battery slurries made of micron-sized NMC, nano-sized CB, and LMW PVDF [8]. The volume fraction of CB, $\varphi_{CB}$, used here was 0.025, which is above the critical fraction for gelation of 0.02 found experimentally. In addition to the volume fraction requirement, colloidal gelation requires three additional criteria. First is the gravitational Peclet number $Pe_g$,

$$Pe_g = \frac{2\pi\Delta\rho g a^4}{9k_B T}\left(\frac{R}{a}\right)^{d_f+1} \tag{1}$$

where $\Delta\rho$ is the difference between solvent and particle density, $g$ is the gravitational constant, $a$ is the particle radius, $R$ is the cluster size, and $d_f$ is the fractal dimension. $R$ is determined by the volume fraction of CB, and $d_f$ is typically between 1.7 and 1.9 for diffusion-limited cluster aggregation [8]. In the case of CB, where primary particles form larger aggregates, $a$ is the dominant aggregate size instead of particle radius. Gelation requires that $Pe_g < 1$, indicating that Brownian diffusion overcomes sedimentation. Our previous work found that the dominant aggregate size of CB in the presence of LMW PVDF is $a$ = 50 nm, resulting in a $Pe_g = 10^{-2}$. Because of the larger size and higher density, micron-sized NMC was shown to play no role in the gelation transition; its $Pe_g \gg 1$, see Table 2.

**Table 2.** Gelation criteria for slurry microstructure.

| Parameter | Component | Expected Value |
|---|---|---|
| Gravitational Peclet number | CB | $1 \times 10^{-2}$ |
| | NMC | $4 \times 10^6$ |
| Polymer concentration ratio | LMW PVDF | 2 |
| | HMW PVDF | 10 |
| Attraction energy | CB & LMW PVDF | $-0.5$ |
| | CB & HMW PVDF | 0.25 |

Next, the ratio of the polymer mass concentration $C_P$ and the critical overlap concentration $C_P^*$ is given by

$$\frac{C_P}{C_P^*} = \frac{C_P 4\pi N a R_g^3}{3Mw} \qquad (2)$$

$C_P^*$ describes the solution concentration at which polymer chains begin to interact with each other. It depends on the radius of gyration $R_g$ as well as the particle radius and molecular weight Mw. The third requirement for gelation is that the depletion interaction potential $-V_C$ must be attractive,

$$-\frac{V_C}{k_B T} = \log\left[12\left(\frac{R_g}{a}\right)^2 \phi_0\right], \qquad (3)$$

where $\phi_0$ is particle volume fraction. $V_C$ depends critically on the ratio of polymer and particle volumes. Because the depletion zone around the particles must create a large enough osmotic pressure to push away the polymer, particles that are significantly smaller than the polymer will not generate a sufficient driving force to induce gelation. Expected values for the parameters given by Equations (1)–(3) are summarized in Table 2. These parameters do not involve mixing effects, but rather define a slurry's propensity to form a gel. CB particles satisfy the $Pe_g$ criteria, which does not depend on the polymer properties, while both LMW and HMW PVDF are close to the $C_P/C_P^*$ criteria and similar to each other. The only distinguishing criteria between HMW and LMW is the attraction energy, Equation (3). A full discussion of the constraints and parameters are discussed in Morelly et al. [8].

The values reported in Table 2 along with several observations in Figure 2 help to explain the vastly different microstructures observed from the same slurry composition. For example, LMW PVDF formulations always lead to a weak or strong gel; no viscoelastic fluid responses were observed. In accordance with colloidal gel literature, there is a correlation between the strength of the gel and the magnitude of shear rate applied during mixing. For example, the highest moduli are observed after mixing with a planetary mixing, which generates the highest shear rates. Eberle et al. show that when the hydrodynamic forces are larger than the interparticle attractive forces, the particle aggregates are broken down and colloidal particles are well dispersed by the flow field [25]. Such highly dispersed particles form more dense networks compared to mixing at low shear rates [26,27]. Similarly, Mayer et al. also recently showed that high shear rates were more effective at reducing the size of CB agglomerates [28]. For LMW PVDF, the distinct difference between overhead and planetary mixing on the magnitude of the modulus is related to the degree of aggregate breakup and particle dispersion before network formation.

On the other hand, HMW PVDF formulations show two extremes: a very strong gel and a viscoelastic fluid. Table 2 suggests that only formulations with LMW PVDF should form a network because the interaction potential between HMW PVDF and particles is repulsive. According to Equation (3), inducing a favorable attraction energy requires the size of the polymer chain to be reduced to a radius of gyration smaller than the radius of the average CB particle. Increasing shear is much more likely to decrease than to increase the size of the CB aggregates [28]; further, larger aggregates would be less likely to satisfy the $Pe_g$ criteria. This leaves only one obvious suggestion for the observed strong gels for HMW PVDF: shear must be causing chain scission during mixing.

In chain scission, the friction of mixing breaks the covalent bonds of polymer chains to decrease the Mw and increase the solution molarity [29–33]. The effect of polymer Mw on the attractive energy $-V_C$ as calculated in Equation (3) can therefore explain the observed phenomena. Figure 3 shows how the Mw of PVDF impacts the interaction potential, assuming that the dominant aggregate size remains the same. The hollow symbols are interaction potentials calculated for Mws of PVDF with reported values of $R_g$ in the literature [34]. The solid symbols represent the Mw of PVDF (380 and 1367, see Methods and Materials) used in this work, where $R_g$ was calculated by fitting $R_g = CMw^{3/5}$ to reported data, where $C$ is a scaling constant and the exponent is that of a good solvent [34,35]. For

HMW PVDF to induce an attractive energy between particles, the Mw would need to be reduced below approximately 1000 kg/mol.

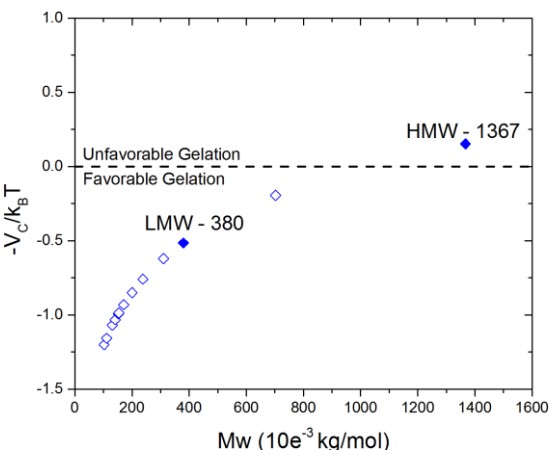

**Figure 3.** Interaction potential between CB particles as a function of polymer Mw. Open symbols represent reference data for $R_g$ of PVDF in NMP. Closed symbols are the expected values of the interaction potential for our starting components, based on manufacturer's data for low- and high-molecular weight (see Methods). HMW PVDF gels despite theoretical predictions, further supporting the occurrence of shear scission.

Steady shear viscosity was used to experimentally confirm the occurrence of polymer scission during the polymer mixing step. Figure 4a shows the steady shear viscosity as a function of shear rate for LMW and HMW PVDF solutions prepared with three different mixers. For a given polymer concentration, viscosity is directly related to the average Mw of the polymer, $\eta \sim Mw^\alpha$, where $\alpha$ is the system-specific power law dependence [36]. The weak dependence of $\eta$ on shear rate for LMW suggests a low value of $C_P/C_P^*$. The higher values of $\eta$ and stronger shear thinning behavior suggest a larger value of $C_P/C_P^*$ as predicted. Solutions of HMW PVDF prepared by the lowest shear method of dissolution method have a much higher viscosity at all shear rates than solutions prepared by the overhead and planetary mixers. This change suggests that shear applied to the polymer during solubilization induces chain scission. The same trend, although much weaker, is observed for the LMW PVDF samples.

In a separate experiment, PVDF was allowed to slowly dissolve without shear before overhead or planetary mixer shear was applied to the polymer solution. Figure 4b shows no changes in the steady shear viscosity after shearing the dissolved polymer solution. The contrast between Figure 4a,b shows that, for the mixing equipment used here, polymer scission occurs as the polymer is blended into solution, but not after it has been solubilized.

If polymer scission can occur during polymer mixing, then it is also likely to occur during particle mixing. Both NMC and CB provide high surface area that can catalyze shear-induced bond breaking, and the use of particulate grinding media has been well-documented previously [37,38]. For example, Ndour et al. showed that the mixing of polymer binder with particles leads to a "deleterious" impact on binder Mw [39]. Consistent with this hypothesis, the gel response of HMW Dissolution-Planetary in Figure 2b in comparison to the fluid response of HMW Dissolution-Overhead strongly suggests that additional polymer scission occurs during particle mixing. In addition, high shear rates have been shown to induce more open network structures that lead to higher moduli [12]. Because a gel is not predicted for the HMW PVDF, and because the polymer of Figure 2b was mixed by the no-shear dissolution method, chain scission during the particle mixing step must be responsible for a decrease in the Mw. Furthermore, the strongest gels (and presumed lowest Mw) formed from HMW PVDF all were formed after the planetary mixer during the particle mixing step.

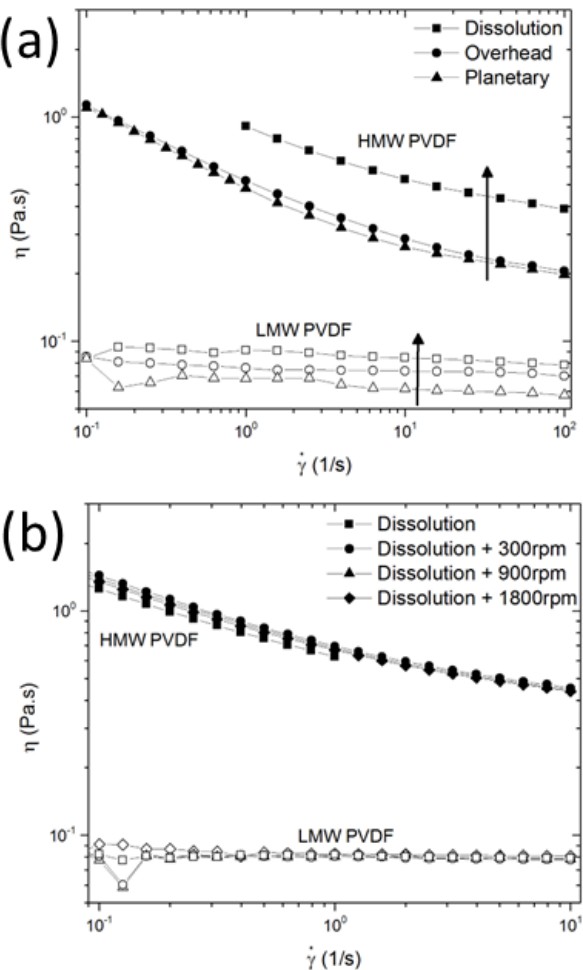

**Figure 4.** Steady shear viscosity as a function of shear rate for PVDF solutions after the polymer mixing step from Figure 1. Hollow and closed symbols correspond to LMW PVDF and HMW PVDF respectively. (**a**) PVDF solutions after being mixed with different mixers. The arrows indicate the direction of higher Mw, higher viscosity, and lower applied shear. (**b**) PVDF solutions after being allowed to dissolve on their own and then mixed in the planetary mixer at different speeds.

Several attempts in this work to measure Mw directly with chromatography were unsuccessful. The presence of carbon and oxide particles in the slurry prevents optical methods of characterization, and separating polymer and particles in a colloidal gel is extremely challenging. By definition, the percolating network resists sedimentation. Ultra-centrifugation might allow separation, but would introduce additional uncharacterized shear. The similarity of $R_g$ and $a$ prevents effective separation via filtration. Additionally, use of PVDF and NMP introduces many materials compatibility challenges for standard chromatography equipment, in contrast to water-soluble systems [40].

Despite these challenges to direct measurement of Mw, our results are consistent with findings by Ndour et al. and Chartrel et al., who measured the impact of ball milling poly(acrylic acid) and carboxymethyl cellulose binders along with silicon and carbon particles [39,40]. Size exclusion chromatography found that the average Mw decreased after ball milling, and that higher Mw binders saw greater changes [40]. Haberzettl et al. also found that carboxymethyl cellulose binders experienced mechanical degradation during slurry mixing [41]. The results here add to the growing body of evidence that shear during slurry mixing can induce polymer scission, including binders as ubiquitous as PVDF.

### 3.2. Effects of Mw on Electrode Reproducibility and Performance

The impact of initial Mw and Mw after shear on electrode performance is determined from twelve slurries, processed into batteries, and tested for rate capability. Four to six electrodes were tested from each electrode film. Despite a similar, homogeneous appearance for all films, electrodes made with the overhead mixer cycled with very low success rates (<50%) and yielding poor statistics for analysis. Previous reports have suggested that viscoelastic fluids are more difficult to process [42], consistent with our results. Therefore, only slurries made with planetary particle mixing (red and green curves in Figure 2) are discussed further. The results of these tests were averaged by Mw and are shown in the form of a semi-log Peukert plot in Figure 5a. Rate capability for individual mixing protocols can be found in Figure S1, and representative discharge curves in Figure S2.

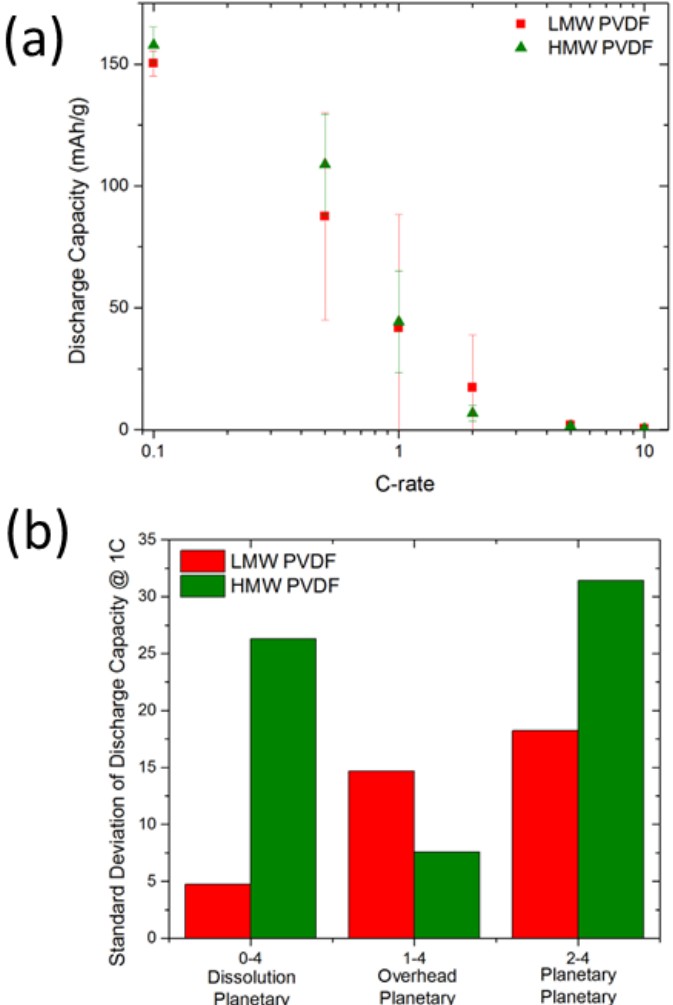

**Figure 5.** (**a**) Average of discharge capacity from six slurries in Figure 2 averaged by starting PVDF molecular weight (**b**) standard deviation for 1C as a function of "total shear" applied during mixing, as explained in the text. Higher Mw and higher shear lead to more variation between coin cell samples, possibly because of greater heterogeneity in binder Mw.

Electrodes made from HMW PVDF slurries have overall a higher performance than those made from LMW PVDF; however, there is significant variation between samples. We previously showed that shear rate during coating and temperature during drying significantly affect battery performance by altering short and long-range carbon-active particle interactions [12]. Overall low-carbon systems such as those here, must have a balance between short and long-range electronic contacts to exhibit reasonable/good

performance [43]. Entwistle defined long-range electronic contacts to have a length scale of 0.1 to 10 um, representing connections along the carbon network, while short-range contacts represented connections between carbon and oxide particles at a length scale of 1–100 nm. Both Stephenson et al. and Albertus et al. found that, because of the short length scale, resistance due to short-range contacts appears not as a bulk electronic conductivity, but rather as an interfacial resistance. While the resistacce is ohmic, not electrochemical, it is therefore mathematically indistinguishable from the linear regime of the Butler Volmer kinetic equation, commonly referred to as charge-transfer resistance [44,45]. The electrode made with HMW PVDF arguably has more favorable carbon-active material interactions (short-range contacts) after coating and drying. One reason for this is that higher Mw reduces the interaction potential between particles and may allow for more rearrangement of the colloidal system during drying. Such rearrangements may allow for the development of better short and long-range order in the HMW PVDF electrodes. More studies are needed at different shear rates and drying temperatures to test this hypothesis. The relationships between carbon, polymer, and active material microstructure and battery performance are extremely complex and the subject of many research efforts across the globe.

While there are differences between HMW and LMW electrodes, Figures 5a and S2 show very high standard deviations, indicating large variation between individual coin cells. This variation is not due to coin cell fabrication methods, which were identical to those in Ref. [43]. We hypothesize that a distribution of binder Mw, generated by polymer scission during shear, may contribute to sample heterogeneity. While the average performance was similar for the different mixing protocols, sample variation was not. Figure 5b shows the standard deviation for the discharge capacity at a discharge rate of 1C as a function of "total" shear. Two numbers are qualitatively assigned to compare the shear applied for each step. The first number indicates the shear from the polymer mixing step with dissolution, overhead and planetary represented by 0, 1 and 2, respectively. The second number indicates the shear from the particle mixing step, with the planetary mixer being 4. Higher standard deviation between samples represents more spatial heterogeneity in the electrode film. A recent study using Raman spectroscopy to quantify electrode inhomogeneity found strong correlations between slurry microstructure and electrode homogeneity [46]. Here, spatial variability may be impacted by inhomogeneity of the PVDF Mw. If Mw affects overall electrode performance by influencing electronic connectivity, porosity, and other microstructural parameters, spatial variation in Mw will generate spatial variation in performance as well. While electrodes made from HMW slurries showed overall higher performance, the standard deviation of the electrodes was also overall higher. High variability of Mw due to polymer scission and poor process control may result in more heterogeneous electrodes that explain this difference in standard deviation.

The results presented here have important implications for reproducibility. Common standard operating procedures for slurry mixing include steps such as "mix for ten minutes, or until polymer is fully dissolved and solution is clear". Such instructions lead to variation in mixing duration between batches. Figure 6a shows the effect of this variation. Three LMW slurries were generated with identical planetary particle mixing, but planetary polymer-mixing time varying from 5 to 20 min. The variation in mixing times results in different amounts of polymer scission and thus different microstructures and degrees of gelation. Figure 4b shows that polymer scission can be avoided in the polymer mixing step by dissolving polymer instead of introducing shear. This process also improves the reproducibility of the slurry microstructure, as shown in Figure 6b. When the LMW PVDF is first dissolved before adding particles, the resulting rheological responses are much more similar.

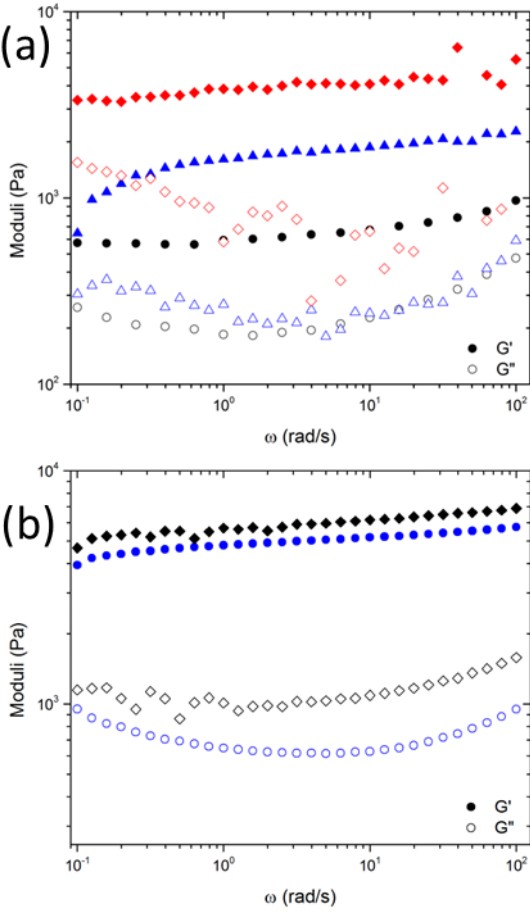

**Figure 6.** Reproducibility of LMW PVDF slurries made with the Planetary − Planetary mixing protocol (**a**) shows three slurries where the only difference was the amount of time during the polymer mixing step. The diamonds, triangles, and circles are in order of decreasing polymer mixing time. (**b**) shows two slurries where LMW PVDF was allowed to dissolve into NMP before the particles were mixed for a set amount of time. Filled symbols represent G′, the loss modulus, and hollow symbols represent the storage modulus G″.

## 4. Conclusions

Systematic studies controlling shear intensity during polymer mixing and particle mixing of battery slurry preparation showed that PVDF binder readily undergoes polymer scission in both mixing steps. Battery slurries made with HMW polymer are not predicted to form colloidal gels, based on the unfavorable interaction potential between polymer and CB aggregates, but gelation still occurs for HMW PVDF when the slurry is mixed under high shear. Steady shear viscosity confirmed that polymer scission occurs during polymer mixing. Comparison of slurry microstructures after identical polymer mixing but variable particle mixing strongly suggests that additional scission occurs during particle mixing. Both processes result in history effects, depending on the exact shear observed during mixing. Electrodes made with HMW PVDF generally outperformed those made with LMW PVDF, but the optimal mixing procedure was obscured by scatter between samples. Increasing standard deviation between electrodes increased with mixing shear, possibly because the lack of control during the polymer scission process leads to local heterogeneities. Polymer scission can be limited to the particle-mixing step and avoided during polymer mixing by first allowing the polymer to dissolve in the absence of shear. Unassisted dissolution of polymer in solution also increases the batch-to-batch reproducibility of the slurry microstructure. Ultimately, our results shed light on an important source of irreproducibility in battery processing. Controlling for sources of error from mixing shear will lead to more efficient process optimization across researchers and laboratories.

**Supplementary Materials:** The following supporting information can be downloaded at: https: //www.mdpi.com/article/10.3390/batteries10020046/s1, Figure S1: Rate capability of electrodes made from slurries discussed in Figure 5; Figure S2: Representative discharge curves.

**Author Contributions:** Conceptualization, S.L.M., N.J.A. and M.T.; methodology, S.L.M.; investigation, S.L.M. and R.M.S.; writing—original draft preparation, S.L.M.; writing—review and editing, N.J.A. and M.T.; supervision, N.J.A. and M.T.; project administration, N.J.A. and M.T.; funding acquisition, N.J.A. and M.T. All authors have read and agreed to the published version of the manuscript.

**Funding:** This research was funded by the National Science Foundation, grant number CBET-1929755.

**Data Availability Statement:** Discharge curves, cycling data, and rheological characterization data are available upon reasonable request to the authors.

**Conflicts of Interest:** The authors declare no conflict of interest. The funders had no role in the design of the study; in the collection, analyses, or interpretation of data; in the writing of the manuscript; or in the decision to publish the results.

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
