# Peer review of "Impact of Mixing Shear on Polymer Binder Molecular Weight and Battery Electrode Reproducibility"

_batteries, doi:10.3390/batteries10020046_

Round 1
Reviewer 1 Report
Comments and Suggestions for Authors
In this manuscript, the authors applies the rheology to understand the impact of shear during mixing and polymer molecular weight on slurry micro-structure and electrode performance. For further improving the manuscript, the concerns should be addressed before its further consideration for publication. The authors could consider the following comments:
1. In the manuscript, the Ni0.33Mn0.33Co0.33O2 (NMC) should be written as the “LiNi0.33Mn0.33Co0.33O2 (NMC)”.
2. It can be seen that four lines are present in Figure 2. What did they represent for, respectively?
3. What conclusions could be drawn from the Figure 5a? Please give a detailed explanation.
4. What is the HMW-1367 and LMW-380 in Figure 3?
5. What is the G' and G'' in Figure 6?
6. The quality of the Figures needs to be further improved.
7. Please supply some documents that published in near two years.
Reviewer 2 Report
Comments and Suggestions for Authors
The paper describes a systematic study on an influenece of the slurry preparation conditions on the performance of the NMC cathodes for Li-Ion. The paper is well written, study is nicely arranged, experimental methods are adequate, results are clearly presented. However, the experimental work described looks incomplete, and some crucial experiments are missing:
1) Authors describe the viscosity measurements on the solutions of PVDF samples prepared by different techniques, and postulate the polymer chain scission phenomenon. To ensure that such scission truly occurs, the MM characteristics of the resulting samples must be confirmed by at least two methods. The authors should study the MM distribution of said PVDF solutions by GPC, light scattering or ultracentrifugation. Otherwise, one of the main conclusions of this work, namely shear-dependent chain scissoring of the PVDF, can't be considered as supported by the experimental results.
2) The main question that was not adressed in this study is the factors determining the decrease of C in the particular samples. Authors have shown the effect of various slurry praparation techniques on the rheology and molecular mass distribution, but what does it means in the electrochemical context? Is the C decrease caused by the degradation of the electrical or ionic conductivity in bulk, or impeded interfacial charge or ion injection, or degradation of the intrinsic charge storage ability of the NMC particles?
I strongly recommend the authors to acqure the electrochemical impedance spectra and/or PITT/GITT profiles of the electrodes, and measure the DC conductance of the electrode materials. Obtained data will allow authors to find the factor which primarily affects the charge storage performance of the materials depending on the slurry preparation technique.
The paper in the present form needs major revision. Once the authors will perform the experiments described above, the paper can be accepted to publication.
Comments on the Quality of English LanguageTypo correction is required. Figure 1 is incorrectly trimmed.
